# Revisiting Spectral Representations in Generative Diffusion Models

## Abstract

Diffusion models have shown remarkable performance on diverse generation tasks. Recent work finds that imposing representation alignment on the hidden states of diffusion networks can both facilitate training convergence and enhance sampling quality, yet the mechanism driving this synergy remains insufficiently understood. In this paper, we investigate the connection between self-supervised spectral representation learning and diffusion generative models through a shared perspective on perturbation kernels. On the diffusion side, samples (e.g., images, videos) are produced by reversing a stochastic noise-injection process specified by Gaussian kernels; on the spectral representation side, spectral embeddings emerge from contrasting positive and negative relations induced by random perturbation kernels. Motivated by this, we propose a self-supervised spectral representation alignment method to facilitate diffusion model training. In addition, we clarify how joint spectral learning can benefit diffusion training from a geometric perspective. Furthermore, we find that the optimization of the spectral alignment objective is in an equivalent form of diffusion score distillation in the representation space. Building on these findings, we integrate a spectral regularizer into diffusion training objectives to improve the performance of diffusion models on multiple datasets. Experiments across images and 3D point clouds show consistent gains in generation quality.

## 1 Introduction

Diffusion models (Sohl-Dickstein et al., 2015; Song & Ermon, 2019; Ho et al., 2020; Song et al., 2021) have demonstrated strong generative capabilities across diverse domains, including images (Rombach et al., 2022; Dhariwal & Nichol, 2021), videos (Brooks et al., 2024; Bao et al., 2024), 3D shapes Nichol et al. (2022); Zhao et al. (2025), molecules (Hoogeboom et al., 2022), etc. Their core idea is to reverse a diffusion process defined by a Gaussian perturbation kernel Song et al. (2021). To achieve this, diffusion models learn to estimate the time-dependent score functions on *perturbed data*. Notably, this learning setup closely mirrors self-supervised representation learning, where models are also trained on data deliberately altered through perturbations or augmentations (HaoChen et al., 2021; Zbontar et al., 2021; Bardes et al., 2022; Sohn, 2016; Oord et al., 2018; Tian et al., 2020). In both cases, performance hinges on extracting useful structure from *perturbed inputs*: self-supervised methods aim to capture universal representations for downstream tasks, while diffusion models are dependent on appropriate representations to recover clean samples for the specific generation task. This parallel motivates a key question: Do diffusion models and self-supervised representation learning share a fundamental connection, and can exploiting it improve generative modeling?

Recent works have begun to explore the link between diffusion models and self-supervised representation learning (Preechakul et al., 2022; Yang et al., 2022; Abstreiter et al., 2021; Mittal et al., 2022). On the one hand, several studies reuse diffusion models as self-supervised representation learners (Chen et al., 2024; Xiang et al., 2023; Mukhopadhyay et al., 2023; Zhang et al., 2022), showing that meaningful features emerge during diffusion training and transfer well to downstream tasks (Tang et al., 2023; Park et al., 2023). On the other hand, REPA (Yu et al., 2024) takes the opposite direction, demonstrating that representation learning can in turn benefit diffusion models. By aligning the hidden states of denoising networks with clean-image embeddings from pretrained encoders such as DINOv2 (Oquab et al., 2023), REPA achieves faster convergence and stronger image

generation. Nevertheless, REPA relies on representations from external foundation models, which are often unavailable for other modalities such as point clouds or graphs. Moreover, the broader intrinsic connection between diffusion and self-supervised learning remains unclear.

In this work, we conduct a pilot study on the synergy between self-supervised representation learning and diffusion-based generative modeling. Specifically, we focus on spectral representation learning (SRL) within self-supervised methods, inspired by prior works that admit multiple effective formulations built from perturbation kernels (HaoChen et al., 2021; Deng et al., 2022a; Pfau et al., 2018). Through the lens of perturbation kernels, we first review and unify the formulations of diffusion models and SRL under a shared stochastic process parameterization (Section 2 and Section 3). Given that spectral representations preserve neighborhood structure on the underlying data manifold (Deng et al., 2022a), it is plausible that incorporating spectral representation into diffusion training can inform the denoising networks of the latent, time-evolving local data geometry, thereby leading to better generative performance. Motivated by this, we propose a novel training strategy for diffusion models that regularizes the diffusion model's intermediate representations to align with the eigenfunctions of a time-varying kernel integral operator defined by a shared diffusion perturbation kernel (Section 4.1). Moreover, we establish a theoretical duality between representation learning and generative modeling (Section 4.2). In particular, we show that optimizing our spectral self-supervised objective is (in gradient) equivalent to diffusion score distillation (Poole et al., 2022) formulated via a KL divergence. This distributional alignment induces mode-seeking dynamics in representation space: embeddings are pulled toward their local data distribution and pushed away from mismatched regions, thereby facilitating the end goal of generative modeling.

Experimentally, our proposed self-supervised spectral representation alignment yields consistent gains in diffusion training for image generation across four datasets with different data diversity, scales, and domains. Moreover, on point-cloud generation where pretrained encoders are unavailable, it attains strong performance over the baseline method, highlighting the method's potential to complex generative settings in which encoder pretraining is impractical.

## 2 REVIEW DIFFUSION MODELS FROM PERTURBATION KERNELS

In diffusion-based generative models (Ho et al., 2020; Song & Ermon, 2019; Song et al., 2021), data samples $\boldsymbol{x}_0 \sim p_{\text{data}}(\boldsymbol{x}_0)$ in $d$-dimensional space ($\boldsymbol{x}_0 \in \mathbb{R}^d$) are first transported to a standard Gaussian distribution by gradually perturbing the original data distribution with random Gaussian noise. Specifically, the perturbation kernel $p_{0t}(\boldsymbol{x}_t|\boldsymbol{x}_0)$ is defined as $\mathcal{N}\left(\boldsymbol{x}_t; s(t)\boldsymbol{x}_0, s(t)^2\sigma(t)^2\boldsymbol{I}\right)$, where $t$ is the timestep of the diffusion process, $s(t)$ is a scaling coefficient, and $\sigma(t)$ is the noise scale at $t$. Given this perturbation kernel, the SDE of the forward process is determined as follows:

$$\mathrm{d}\boldsymbol{x} = f(t)\boldsymbol{x}\,\mathrm{d}t + g(t)\mathrm{d}\boldsymbol{w}_t, \tag{1}$$

where $f(t)\boldsymbol{x}$ is a drift term, $g(t) : \mathbb{R} \to \mathbb{R}$ is the diffusion coefficient of $\boldsymbol{x}$, and $\boldsymbol{w}_t$ is the standard Wiener process. The following equations describe the relations between $f(t)$, $g(t)$, $s(t)$, and $\sigma(t)$, which illustrate how the SDE can be derived from the perturbation kernel (Karras et al., 2022):

$$f(t) = \dot{s}(t)/s(t) \quad g(t) = s(t)\sqrt{2\dot{\sigma}(t)\sigma(t)}. \tag{2}$$

Conversely, the scaling and noise scale terms in the perturbation kernel $p_{0t}$ can be rewritten with respect to $f(t)$ and $g(t)$:

$$s(t) = \exp\left(\int_0^t f(\xi)\mathrm{d}\xi\right), \quad \sigma(t) = \sqrt{\int_0^t \frac{g(\xi)^2}{s(\xi)^2}\mathrm{d}\xi}. \tag{3}$$

To sample the original data distribution from a randomly sampled noise, we can reverse the diffusion process. As introduced in the literature (Song et al., 2021), the reverse process of Equation 1 can be described as the SDE below:

$$\mathrm{d}\boldsymbol{x} = \left[\boldsymbol{f}_t(\boldsymbol{x}) - g(t)^2\nabla_{\boldsymbol{x}}\log p_t(\boldsymbol{x})\right]\mathrm{d}t + g(t)\mathrm{d}\boldsymbol{w}_t, \tag{4}$$

where $p_t(\boldsymbol{x})$ is the perturbed data distribution evolving over the process time-dependently, and $\nabla_{\boldsymbol{x}}\log p_t(\boldsymbol{x})$ is a score function which can be estimated by training deep neural networks $\boldsymbol{s}_\phi$ to

match the true scores:

$$\mathcal{L}_{\mathrm{diff}}(\theta) = \mathbb{E}_t \left[ \omega(t) \mathbb{E}_{\boldsymbol{x}_0 \sim p_{\mathrm{data}}, \, \boldsymbol{x}_t \sim p_{0t}(\boldsymbol{x}_t | \boldsymbol{x}_0)} \left[ \| \boldsymbol{s_\theta}(\boldsymbol{x}_t, t) - \nabla_{\boldsymbol{x}_t} \log p_{0t}(\boldsymbol{x}_t | \boldsymbol{x}_0) \|_2^2 \right] \right] \tag{5}$$

$$= \mathbb{E}_t \left[ \omega(t) \mathbb{E}_{\boldsymbol{x}_0 \sim p_{\mathrm{data}}, \, \boldsymbol{x}_t \sim p_{0t}(\boldsymbol{x}_t | \boldsymbol{x}_0)} \left[ \left\| \boldsymbol{s_\phi}(\boldsymbol{x}_t, t) + \frac{\boldsymbol{x}_t - s(t)\boldsymbol{x}_0}{s(t)^2 \sigma(t)^2} \right\|_2^2 \right] \right], \tag{6}$$

where $\omega(t)$ is a time-dependent re-weighting of score-matching losses across different $t$. Formulating diffusion processes with perturbation kernels facilitates score matching in the two aspects: 1) Given $\boldsymbol{x}_0$ and $\boldsymbol{x}_t$, the true scores have analytic expressions. 2) The perturbation kernel $p_{0t}$ allows for a "simulation-free" forward process, i.e., one can sample $\boldsymbol{x}_t = s(t)\boldsymbol{x}_0 + s(t)\sigma(t)\boldsymbol{\epsilon}$ without numerically simulating the SDE in Equation 1. Moreover, flow-based diffusion models (Liu et al., 2022) can be defined by perturbation kernels as well (see Appendix A for the derivation).

## 3 SPECTRAL REPRESENTATION FROM PERTURBATION KERNELS

In this section, we will revisit a family of self-supervised learning approach that restores data representations in the spectral domain of kernels, a.k.a spectral representation learning (SRL). In particular, we examine Neural Eigenmap (Deng et al., 2022a), which trains a neural network to approximate the principal eigenfunctions of a kernel integral operator. Solving the resulting eigenvalue problem then yields representations in the eigenspace. Given a kernel $\kappa(\boldsymbol{x}, \boldsymbol{x}')$, the corresponding kernel integral operator can be defined as:

$$(\mathcal{T}_\kappa h)(\boldsymbol{x}) = \int \kappa(\boldsymbol{x}, \boldsymbol{x}') h(\boldsymbol{x}') p(\boldsymbol{x}') d\boldsymbol{x}', \tag{7}$$

where $f \in L^2(\mathcal{X}, p)$, i.e., $f$ is a square-integrable function w.r.t $p$. $\mathcal{X}$ is a support, and $p$ is a probability distribution defined over the support. Intuitively, this operator can be understood as the continuous-domain analogue of matrix multiplication. Here, we consider this type of kernel:

$$\kappa(\boldsymbol{x}, \boldsymbol{x}') = \frac{p(\boldsymbol{x}, \boldsymbol{x}')}{p(\boldsymbol{x})p(\boldsymbol{x}')}, \quad p(\boldsymbol{x}, \boldsymbol{x}') = \mathbb{E}_{\bar{\boldsymbol{x}} \sim p_{\mathrm{data}}}[p(\boldsymbol{x}|\bar{\boldsymbol{x}})p(\boldsymbol{x}'|\bar{\boldsymbol{x}})], \tag{8}$$

where $p_{\mathrm{data}}$ is a clean data distribution, $p(\boldsymbol{x}|\bar{\boldsymbol{x}})$ is a data perturbation kernel. Following NeuralEF (Deng et al., 2022b), Neural Eigenmap reformulates the eigenfunction problem of $\mathcal{T}_\kappa \psi^j = \mu \psi^j$ into an optimization problem:

$$\max_{\psi_j} R_{j,j} - \alpha \sum_{i=1}^{j-1} R_{i,j}^2, \quad \text{for } j = 1, .., K, \tag{9}$$

$$R = \mathbb{E}_{p(\boldsymbol{x}, \boldsymbol{x}')} \left[ \psi(\boldsymbol{x})\psi(\boldsymbol{x}')^\top \right] \approx \frac{1}{B} \sum_{b=1}^{B} \psi(\boldsymbol{x}_b)\psi(\boldsymbol{x}'_b)^\top, \tag{10}$$

where $K$ is the number of eigenfunctions, $\psi(\boldsymbol{x}) = \left[ \psi^1(\boldsymbol{x}), ..., \psi^K(\boldsymbol{x}) \right] \in \mathbb{R}^K$ denotes the vector comprising the first $K$ eigenfunctions evaluated at $\boldsymbol{x}$, $B$ is the number of data samples, $\boldsymbol{x}_b$ and $\boldsymbol{x}'_b$ are independently sampled from the perturbation kernel $p(\boldsymbol{x}|\bar{\boldsymbol{x}}_b)$ conducted on the same clean data $\bar{\boldsymbol{x}}_b$. We can parameterize $\psi$ by a neural network, and the network parameters $\theta$ can be optimized through the following loss function:

$$\mathcal{L}_{ef}(\theta) = -\sum_{j=1}^{K} \left( \psi_\theta(\boldsymbol{X}_B)\psi_\theta(\boldsymbol{X}'_B)^\top \right)_{j,j} + \alpha \sum_{j=1}^{K} \sum_{i=1}^{j-1} \left( \mathrm{sg}(\psi_\theta(\boldsymbol{X}_B))\psi_\theta(\boldsymbol{X}'_B)^\top \right)_{i,j}^2, \tag{11}$$

where $\mathrm{sg}(\cdot)$ denotes stop-gradient operator that converts its argument as an constant with zero derivative, $\alpha$ is the coefficient weighting the regularization applied to the upper-triangular elements, $\boldsymbol{X}_B = [\boldsymbol{x}_1, ..., \boldsymbol{x}_B]$, $\boldsymbol{X}'_B = [\boldsymbol{x}'_1, ..., \boldsymbol{x}'_B]$ are batched input data, $\boldsymbol{x}_b$ and $\boldsymbol{x}'_b$ are perturbed from the same clean data $\bar{\boldsymbol{x}}_b$ for $b = 1, ..., B$, and $B$ is the batch size for mini-batch training. Thereby $\psi_\theta(\boldsymbol{X}_B)$ is a $K \times B$ matrix with the element at $j$-th row, $b$-th column representing the $j$-th eigenfunction evaluated at the $b$-th data sample in the training batch.

The loss function in Equation 11 bears a strong resemblance to other contrastive representation learning objectives (Li et al., 2022; Zbontar et al., 2021). This connection offers a compelling interpretation: data representations can be encoded through the eigenfunctions of a kernel integral operator. Specifically, in the Neural Eigenmap framework, the kernel is constructed from positive pairs obtained via data perturbation, while negative relations among samples perturbed from different clean data points are implicitly imposed as orthogonality regularization of eigenfunctions. Those associated eigenfunctions span a low-dimensional subspace that captures the intrinsic geometry of the data distribution (Coifman & Lafon, 2006).

The perturbation kernels $p(\boldsymbol{x}|\bar{\boldsymbol{x}})$ used for SRL are usually designed as composed data augmentations. For instance, for representation learning on images, $p(\boldsymbol{x}|\bar{\boldsymbol{x}})$ can be a composition of image manipulations, such as color jittering, random flip, Gaussian blur, etc. However, there is no restriction for defining $p(\boldsymbol{x}|\bar{\boldsymbol{x}})$. To study the synergy of SRL and diffusion models, we adopt the same perturbation kernel in diffusion models, i.e, $p_{0t}(\boldsymbol{x}_t|\boldsymbol{x}_0)$. Therefore, once the SDE of a diffusion process is given, a time-dependent perturbation kernel is also determined for SRL.

# 4 BRIDGING SPECTRAL REPRESENTATIONS AND DIFFUSION MODELS

We have reviewed diffusion models and spectral representations through the lens of perturbation kernels. Motivated by their shared principle of learning from perturbed data, we further develop their connection. First, we reformulate Neural Eigenmap within the diffusion framework and integrate spectral representations as a joint training objective to enable self-supervised representation alignment during diffusion model training. Second, we show that spectral representation regularization in our proposed training objective can be interpreted as a special case of diffusion score distillation (Poole et al., 2022; Zhou et al., 2024).

## 4.1 NEURAL EIGENMAP REGULARIZER WITH DIFFUSION PERTURBATION KERNELS

Following Yu et al. (2024), we incorporate SRL as a regularizer within diffusion training, providing self-supervised representation alignment to enhance sampling quality. To establish compatibility between the two objectives, we first recast spectral learning in terms of the diffusion perturbation kernel $p_{0t}(\boldsymbol{x}_t|\boldsymbol{x}_0) = \mathcal{N}(\boldsymbol{x}_t; (1-t)\boldsymbol{x}_0, t\boldsymbol{I})$ (the one used in rectified flow), where $\boldsymbol{x}_0$ is a clean data sampled from $p_{\text{data}}$. Note that our subsequent analysis is insensitive to the specific parameterization of the perturbation kernel; the particular choices of $s(t)$ and $\sigma(t)$ for $p_{\text{data}}$ will not affect our following discussion. Then, a time-varying normalized joint kernel can be defined as follows:

$$\kappa_t(\boldsymbol{x}, \boldsymbol{x}') = \frac{p_t(\boldsymbol{x}, \boldsymbol{x}')}{p_t(\boldsymbol{x})p_t(\boldsymbol{x}')}, \quad p_t(\boldsymbol{x}, \boldsymbol{x}') = \mathbb{E}_{\boldsymbol{x}_0 \sim p_{\text{data}}}[p_{0t}(\boldsymbol{x}_t|\boldsymbol{x}_0)p_{0t}(\boldsymbol{x}'_t|\boldsymbol{x}_0)]. \quad (12)$$

Using this kernel, we further construct its time-varying kernel integral operator $\mathcal{K}_t$:

$$(\mathcal{K}_t h)(\boldsymbol{x}) = \int \kappa_t(\boldsymbol{x}, \boldsymbol{x}')h(\boldsymbol{x}')p_t(\boldsymbol{x}')d\boldsymbol{x}'. \quad (13)$$

Since this operator is time-varying, its eigenfunctions also need to be formulated in a time-dependent manner: $\mathcal{K}_t\psi_\theta^j(\boldsymbol{x}_t, t) = \mu_t\psi_\theta^j(\boldsymbol{x}_t, t)$. By putting the the most $K$ primary eigenfunctions into vectors: $[\psi_\theta^1(\boldsymbol{x}_t, t), \cdots, \psi_\theta^K(\boldsymbol{x}_t, t)]$, time-varying embeddings are obtained as "multi-scale" representations, learned for data under different levels of noise. Plugging $\kappa_t$ into Neural Eigenmap, we can solve the eigenfunction problem using the following spectral loss:

$$\mathcal{L}_s(\theta) = \mathbb{E}_{\substack{t, \ \boldsymbol{x}_0 \sim p_{\text{data}} \\ \boldsymbol{x}_t, \boldsymbol{x}'_t \sim p_{0t}(\boldsymbol{x}_t|\boldsymbol{x}_0)}} \left[ -\text{Tr}\left(\psi_\theta(\boldsymbol{x}_t, t)\psi_\theta(\boldsymbol{x}'_t, t)^\top\right) + \alpha \sum_{j=1}^{K}\sum_{i=1}^{j-1}\left(\text{sg}\left(\psi_\theta(\boldsymbol{x}_t, t)\right)\psi_\theta(\boldsymbol{x}'_t, t)^\top\right)_{i,j}^2 \right], \quad (14)$$

where $t \in (0, 1]$ is a randomly sampled time step, $\boldsymbol{x}_t$ and $\boldsymbol{x}'_t$ are two i.i.d perturbed views of the same clean data samples $\boldsymbol{x}_0$, and the time-conditioned neural network $\psi_\theta(\boldsymbol{x}_t, t)$ parameterizes the eigenfunctions of $\mathcal{K}_t$. Comparing $\mathcal{L}_s$ and $\mathcal{L}_{\text{diff}}$ in Equation 6, both involve sampling random time steps $t$ and perturbed data $\boldsymbol{x}_t \sim p_{0t}(\boldsymbol{x}_t|\boldsymbol{x}_0)$, whereas Equation 14 additionally requires an independently sampled $\boldsymbol{x}'_t$. This permits a practical implementation that jointly optimizes the diffusion and spectral objectives while reusing the same perturbed input, leading to our final training objective:

$$\mathcal{L}(\theta, \phi) = \mathcal{L}_{\text{diff}}(\phi) + \lambda \mathcal{L}_s(\theta), \quad (15)$$

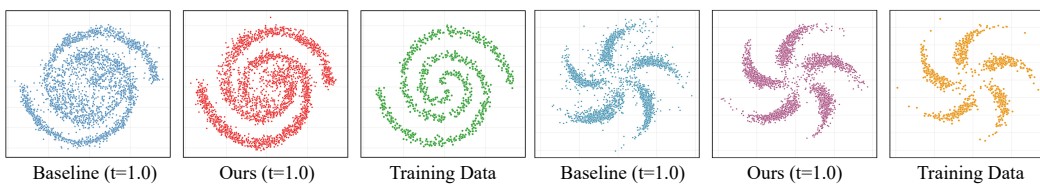

| Baseline (t=1.0) | Ours (t=1.0) | Training Data | Baseline (t=1.0) | Ours (t=1.0) | Training Data |

Figure 1: **Results on synthetic 2D data distributions.** Our method produces a cleaner, more compact sample distribution than the baseline, with fewer outliers.

where $\theta$ denotes parameters of the spectral learner $\psi_\theta$, $\phi$ is a set of parameters of diffusion networks, and $\lambda$ is the coefficient controlling the strength of the spectral regularization.

**Implementation Details.** We follow the implementation of representation alignment in REPA:

- Given input $\mathbf{x}_t$ and $\mathbf{x}'_t$, extract their intermediate hidden states from a chosen layer of the diffusion network as the features to align.

- Feed these states to a shared projection head $P_{\theta'}$ ($\theta' \subset \theta$) to obtain $\psi_\theta(\mathbf{x}_t, t)$ and $\psi_\theta(\mathbf{x}'_t, t)$.

- Evaluate $\mathcal{L}_s$ at $\psi_\theta(\mathbf{x}_t, t)$ and $\psi_\theta(\mathbf{x}'_t, t)$, and back-propagate its gradients to update both the diffusion parameters $\phi$ and the spectral learner parameters $\theta$.

The projection head is a two-layer MLP. We condition it on the timestep, identical to the time modulation in (Peebles & Xie, 2023). We apply L2-BN at the final layer to enforce a normalization constraint on the estimated eigenfunctions (Deng et al., 2022b). To stabilize training, we also normalize each output embedding to bound its magnitude.

**Geometric Interpretation.** The time-dependent embeddings defined by the learned eigenfunctions preserve the local geometry of data points on a latent, time-evolving manifold. This follows the classical spectral paradigm: in algorithms such as spectral clustering (Ng et al., 2001; Shi & Malik, 2000) and diffusion maps (Coifman & Lafon, 2006; Coifman et al., 2005; Nadler et al., 2005), eigenspace embeddings of constructed kernel operators yield coordinates that respect neighborhood structure and facilitate unsupervised clustering. In our setting, the kernel operator $\mathcal{K}_t$ varies with time via the SDE-defined perturbation (Marshall & Hirn, 2018), and the embeddings $\psi_\theta(\mathbf{x}_t, t)$ track the local geometry as it evolves following the diffusion process. In Appendix B, we show that Euclidean distances in the time-dependent eigenspace of $\mathcal{K}_t$ approximate the time-varying diffusion distance (Coifman et al., 2005). This yields multi-scale representations that reflect the intrinsic geometry at each $t$: for small $t$, data remain well separated, so only nearby points have small embedding distances; as $t$ increases and noise dominates, eigenspace distances progressively collapse and become less discriminative.

In Figure 1, we evaluate on two synthetic 2D distributions using simple MLPs trained either with a vanilla diffusion loss or with our spectral regularizer. Our method yields cleaner, more compact samples with markedly fewer out-of-distribution points. On the "2-spirals" data, in particular, it recovers the fine spiral geometry that the baseline misses. These results illustrate that our proposed method can better capture the underlying data geometry prior.

### 4.2 Spectral Representation Learning as Diffusion Score Distillation

We further look into the self-supervised learning objective in Equation 14. Unlike sample-contrastive methods (e.g., HaoChen et al. (2021)), Eq. 14 does not explicitly construct negative examples. Consequently, the spectral regularizer belongs to the dimension-contrastive family in Garrido et al. (2022), which is provably dual to sample-contrastive learning with positives and negatives. From this viewpoint, for a given perturbed sample as an anchor, instances perturbed from different clean examples can be interpreted as negatives, whereas instances perturbed from the same clean example play the role of positives. Interestingly, in its dual (sample-contrastive) form, our spectral regularizer admits a reformulation as diffusion score distillation (Poole et al., 2022).

**Proposition 4.1.** *Minimizing the self-supervised learning objective in Equation 14 via a gradient-based optimizer is equivalent to minimizing the KL divergence $D_{KL}(p_t^{\psi_\theta}(\boldsymbol{x}_t) \,\|\, p_+)$, as the following*

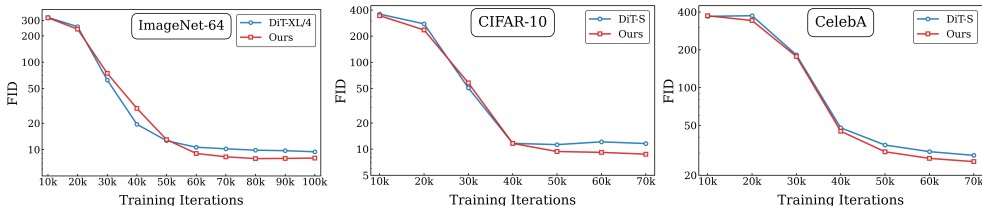

Figure 2: **Visualization of Training Progress.** We plot FID against training iterations for three datasets. These results suggest that our representation learning strategy sustains effective optimization and mitigates the mid-training stagnation observed in the baseline.

| Dataset | Model | Metric | | | | |
|---|---|---|---|---|---|---|
| | | FID ($\downarrow$) | sFID ($\downarrow$) | IS ($\uparrow$) | Precision ($\uparrow$) | Recall ($\uparrow$) |
| ImageNet (res. = 64) | DiT-L/4 baseline | 9.441 | 7.653 | **102.069** | **0.871** | 0.393 |
| | Ours (DiT-L/4) | **7.994** | **7.372** | 78.366 | 0.858 | **0.397** |
| ImageNet (res. = 256, latent) | DiT-XL/2 baseline | 2.508 | 5.630 | 247.891 | 0.822 | 0.566 |
| | REPA (DiT-XL/2) | **1.745** | **5.459** | **296.726** | 0.807 | **0.615** |
| | Ours (DiT-XL/2) | 2.298 | 5.510 | 257.741 | **0.824** | 0.570 |
| CIFAR10 (res. = 32) | DiT-S/2 baseline | 11.588 | 10.680 | 9.042 | 0.719 | 0.384 |
| | Ours (DiT-S/2) | **8.742** | **6.836** | **9.174** | **0.735** | **0.405** |
| CelebA (res. = 32) | DiT-S/2 baseline | 28.806 | 20.569 | **3.431** | 0.685 | 0.453 |
| | Ours (DiT-S/2) | **25.678** | **20.061** | 3.388 | **0.702** | **0.472** |
| FFHQ (res. = 64, uncond.) | DiT-S/2 baseline | 13.766 | 21.982 | 2.997 | 0.731 | 0.331 |
| | Ours (DiT-S/2) | **13.074** | **21.915** | **2.998** | **0.737** | **0.340** |

Table 1: **Evaluation of image generation across four datasets**, with image resolutions and model sizes adapted accordingly. We report FID as the primary metric, and sFID, Inception Scores, Precision/Recall as secondary metrics.

*identity shows:*

$$\frac{\partial \mathcal{L}_s}{\partial \theta} = \mathbb{E}_{\boldsymbol{x} \sim p_t} \left[ \left( \nabla_\theta \psi_\theta(\boldsymbol{x}, t) \right)^\top \nabla_{\psi_\theta(\boldsymbol{x}, t)} \mathcal{L}_s \right] \equiv \nabla_\theta D_{KL}(p_t^{\psi_\theta} \parallel p_+) \tag{16}$$

*where $\nabla_{\boldsymbol{x}_t} \log p_t^{\psi_\theta}$ is equal to the closed-form diffusion scores (Scarvelis et al., 2023) evaluated over negative samples, and the target score $\nabla_{\boldsymbol{x}_t} \log p_+$ matches the closed-form diffusion scores evaluated over positive samples.*

Complete steps to show the above proposition are provided in Appendix C. Intuitively, this KL term measures, at the anchor representation $\psi_\theta(\mathbf{x}_t)$, the discrepancy between a distribution of negative samples and a distribution of positive samples. Since our spectral regularizer applies a stop-gradient to the negatives, minimizing $D_{\text{KL}}\big(p_t^{\psi_\theta} \parallel p_+\big)$ updates $\theta$ so that the anchor $\psi_\theta(\mathbf{x}_t)$ moves to reconcile the score fields of the positive and negative distributions. The resulting dynamics are mode-seeking in representation space, tightening clusters of similar samples while pushing dissimilar ones apart.

## 5 EXPERIMENTS

To validate the effectiveness of guiding diffusion model training via the spectral representation regularization, we conduct experiments on both image (Section 5.1) and point cloud (Section 5.2) generation to validate our proposed method.

### 5.1 IMAGE GENERATION

**Dataset.** We test our method on CIFAR10 (Krizhevsky et al., 2009), CelebA (Liu et al., 2015), FFHQ (Karras et al., 2019), ImageNet (Deng et al., 2009) datasets, which are standard datasets used for training image generation with different data diversity, domain, and scale. For CIFAR10 and CelebA datasets, we resize images into $32 \times 32$ resolution. While for FFHQ, images are resized to $64 \times 64$. For ImageNet, we resize images to two different resolutions: $64 \times 64$ and $256 \times 256$. For

ImageNet $256 \times 256$ experiments, each image is further encoded to $32 \times 32 \times 4$ latents using Stable Diffusion VAE (Rombach et al., 2022), and latent diffusion models are trained on those encoded latents. For other image generation tasks, we conduct diffusion model training on pixel space.

**Training details.** We use DiT (Peebles & Xie, 2023) as the base model and employ the parameterization and training objective of rectified flow Liu et al. (2022). For small datasets (CIFAR-10, CelebA, FFHQ), to mitigate overfitting, we train a small DiT (S, 13M parameters) and patchify images into $2 \times 2$ pixel patches (patch size 2). For ImageNet $64 \times 64$ experiment (models work in pixel space), we train an L/4 model (558M parameters, patch size 4). For ImageNet $256 \times 256$ experiment (models work in latent space), we follow the XL/2 configuration of Peebles & Xie (2023), yielding a 681M-parameter model. Training schedules are adjusted to the dataset scales: S/2 models on CIFAR-10, CelebA, and FFHQ are trained and evaluated at 70k iterations; ImageNet $64 \times 64$ models are trained and evaluated at 100k iterations; and the latent ImageNet $256 \times 256$ model is trained and evaluated at 400k iterations. Since our spectral regularizer requires an additional batch of perturbed samples, we halve the base batch size so that each optimizer step processes the same total number of training examples.

**Evaluation protocol and baselines.** We evaluate generation quality using Fréchet Inception Distance (FID) as the primary metric, complemented by sFID, Inception Score (IS), and the precision/recall pair as secondary measures. All the reported metrics are measured on EMA checkpoints. For pixel-space diffusion, we compare against a vanilla DiT baseline trained under the same setting with ours except no use of our proposed representation learning loss. To further understand the effectiveness of our proposed method, for latent diffusion, we also compare against REPA (Yu et al., 2024), a leading representation-alignment method that leverages encoders pretrained on large-scale external data, which serves as the upper bound of performance. We employ Euler ODE for pixel-space generation and SDE Euler-Maruyama sampler for latent-space generation.

**Results.** As shown in Table 1, using our proposed method for representation learning significantly improves model performance compared to baselines. These performance gains are consistent across different datasets, image resolutions, model scales, and whether the diffusion model applies to pixel or latent spaces. In detail, our method improves FID by 1.5 (15% relatively) on ImageNet with DiT-L/4, 0.2 (8% relatively) on ImageNet with DiT-XL/2, 2.8 (25% relatively) on CIFAR10, 3.1 (11% relatively) on CelebA, and 0.7 (5% relatively) on FFHQ. For latent-space generation, REPA attains the best results, while our method ranks between the baseline and REPA without using any external pretrained encoder. We also include the evaluation results at different training stages. As shown in Figure 2, our method achieves consistently better performance in the second half of training.

## 5.2 POINT CLOUD GENERATION

**Dataset.** Following prior work (Yang et al., 2019; Mo et al., 2023), we use the ShapeNet (Chang et al., 2015) Chair, Airplane, and Car categories with the same preprocessing and data split as Yang et al. (2019). We sample 2,048 points for each shape instance.

**Training details.** For each subset, we use DiT-3D model (Mo et al., 2023) as the base model, which employs 3D window attention in transformer blocks. As the dataset of 3D shapes is relatively small, we use the S/4 configuration (33M parameters, patch size 4). We train the models on each shape category for 10k iterations. We use the same batch-size scheme as in the image-generation experiments.

**Evaluation protocol and baseline.** We follow DiT-3D to evaluate the generated samples with 1-nearest neighbor accuracy (1-NNA) and generated sample coverage (COV). To evaluate 1-NNA, we combine the generated and real samples, use chamfer distance (CD) or earth mover's distance (EMD) to retrieve the most similar sample for each generated sample, and calculate binary classification accuracy of whether the retrieved sample is generated or real (the lower the better). To calculate COV, for each generated shape, we use CD or EMD to retrieve its nearest neighbor in the real data set. After finishing calculation of all generated shapes, we measure the ratio of real reference shape got matched to measure generation diversity (the higher the better).

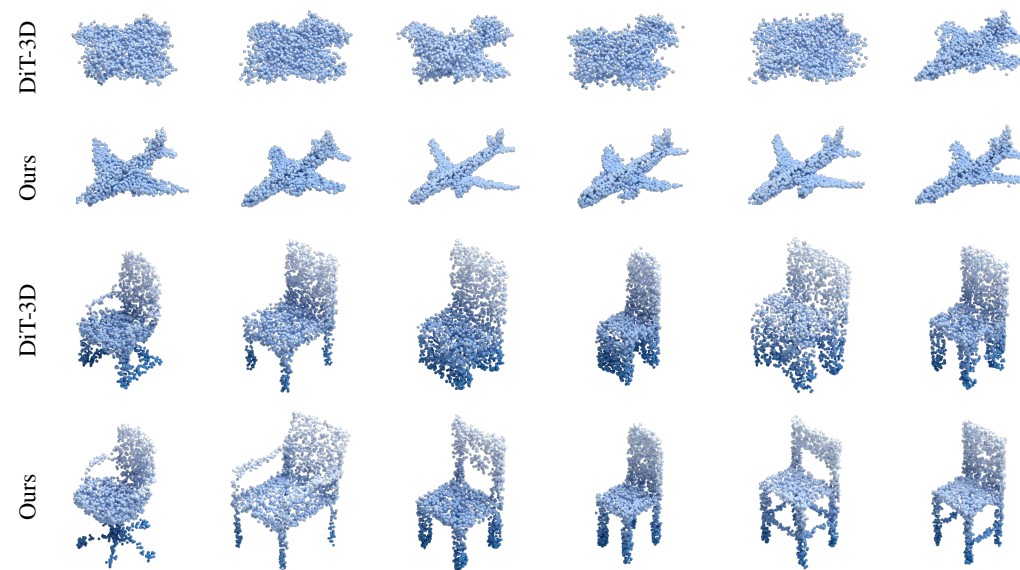

Figure 3: **Visualization of point cloud generation results.** We include generated samples on airplanes (top two rows, generated when model trained with 3K iterations) and chairs (bottom two rows, generated when model trained with 5k iterations).

| Dataset | Iteration | Model | 1-NNA (↓) | | COV (↑) | |
|---|---|---|---|---|---|---|
| | | | CD | EMD | CD | EMD |
| Chair | 5K | DiT 3D-S/4 baseline | 0.850 | 0.875 | 0.295 | 0.221 |
| | | Ours (DiT 3D-S/4) | 0.583 | 0.627 | 0.488 | 0.493 |
| | 10K | DiT 3D-S/4 baseline | 0.565 | 0.545 | 0.504 | 0.511 |
| | | Ours (DiT 3D-S/4) | 0.520 | 0.527 | 0.517 | 0.543 |
| Airplane | 5K | DiT 3D-S/4 baseline | 0.714 | 0.668 | 0.522 | 0.519 |
| | | Ours (DiT 3D-S/4) | 0.601 | 0.556 | 0.561 | 0.523 |
| | 10K | DiT 3D-S/4 baseline | 0.852 | 0.785 | 0.397 | 0.389 |
| | | Ours (DiT 3D-S/4) | 0.607 | 0.562 | 0.570 | 0.600 |
| Car | 5K | DiT 3D-S/4 baseline | 0.788 | 0.738 | 0.378 | 0.482 |
| | | Ours (DiT 3D-S/4) | 0.605 | 0.586 | 0.458 | 0.549 |
| | 10K | DiT 3D-S/4 baseline | 0.730 | 0.682 | 0.427 | 0.427 |
| | | Ours (DiT 3D-S/4) | 0.582 | 0.500 | 0.505 | 0.573 |

Table 2: **Evaluation of 3D point cloud generation** on three subsets of ShapeNet objects. We include 1-NNA and COV computed by either using chamfer distance (CD) or earth mover's distance (EMD) as the criterion for shape retrieval.

**Qualitative results.** Figure 3 shows comparisons between generated point clouds of our method and the baseline in "Car" and "Airplane" categories. Our method demonstrates significantly faster convergence compared to DiT-3D. At an early training stage (3k iterations for airplanes and 5k iterations for chairs), the generations from DiT-3D remain noisy and fragmented, producing messy point distributions without clear geometric structure. In contrast, our approach already produces compact and coherent point clouds that exhibit well-defined shapes with fine-grained details.

**Quantitative results.** Table 2 presents the point cloud generation evaluation results, where our method consistently demonstrates both faster convergence and superior final performance compared to the DiT 3D-S/4 baseline. Notably, after only 5K iterations, our approach already achieves substantial improvements across all datasets. For instance, on the Chair dataset, the 1-NNA (CD/EMD) drops from 0.850/0.875 to 0.583/0.627 (31% and 28% relative improvement, respectively), while the COV (CD/EMD) rises from 0.295/0.221 to 0.488/0.493 (65% and 123% relative improvement,

respectively). Similar trends are observed for Airplane and Car, where our model attains a lower 1-NNA and a higher COV at the early stage of training, highlighting its ability to converge more rapidly. With longer training, our method further improves upon these gains, achieving the best overall results across all metrics. These results clearly indicate that our approach not only converges faster with fewer iterations but also achieves better quality and diversity of generated shapes upon full convergence.

## 6 RELATED WORK

**Improving Representations in Diffusion Models.** Recent work strengthens diffusion by enhancing internal representations. REPA (Yu et al., 2024) aligns denoiser features to pretrained vision encoders (e.g., DINOv2), accelerating convergence and improving sample quality. Its extensions include U-REPA for U-Nets (Tian et al., 2025), REPA-E for joint VAE training (Leng et al., 2025), VideoREPA for video (Zhang et al., 2025), and VAE-side alignment (Yao et al., 2025). REG (Wu et al., 2025) introduces a global semantic token to mitigate the lack of alignment at test time, and HASTE (Wang et al., 2025) adds holistic representation/attention alignment with an alignment-termination criterion to further speed training. However, these approaches assume access to strong foundation encoders, an assumption often violated in resource-constrained domains (e.g., 3D shapes, proteins). Relatedly, You et al. (2023) leverages small-scale category labels, incurring additional annotation cost.

A more relevant line of work builds on the connection between **self-supervised representation learning** and diffusion models. Early works in this direction aim to understand the internal representations of self-supervised diffusion models (Park et al., 2023; Preechakul et al., 2022; Mittal et al., 2022; Chen et al., 2024; Xiang et al., 2023; Mukhopadhyay et al., 2023; Hudson et al., 2024; Li et al., 2025). They show that hidden activations in different time steps encode semantically meaningful information that can be linearly manipulated for image editing and analysis (Park et al., 2023; Tang et al., 2023). Stoica et al. (2025) applies contrastive learning on flow trajectories, improving the uniqueness of flows. A concurrent study (Wang & He, 2025) introduces a dispersive loss that encourages internal representations of different samples to spread apart. While empirically effective, this advance offers primarily an intuitive, self-supervised rationale for improving diffusion models.

**Self-supervised representation learning.** Contrastive learning has emerged as a dominant paradigm for self-supervised visual representation learning (HaoChen et al., 2021; Wang & Isola, 2020; Tian et al., 2020). Early frameworks such as SimCLR (Chen et al., 2020) and MoCo (He et al., 2020; Chen et al., 2021) establish the importance of instance discrimination with large-scale negative sampling. Subsequent works remove the need for negatives, including BYOL (Grill et al., 2020) and SimSiam (Chen & He, 2021), showing that representation quality can emerge purely from positive-pair consistency. Other approaches reformulate contrastive learning through clustering and redundancy reduction, such as SwAV (Caron et al., 2020), Barlow Twins (Zbontar et al., 2021), and VICReg (Bardes et al., 2022). More recently, DINO (Caron et al., 2021; Oquab et al., 2023; Siméoni et al., 2025) advanced self-distillation with vision transformers, producing strong transferable features that have become standard teachers for aligning diffusion models. Collectively, these methods provide the foundation for self-supervised representation alignment in generative models.

## 7 CONCLUSION

In this work, we investigate the connection between self-supervised spectral representation learning and diffusion models through the shared lens of perturbation kernels. Leveraging this alignment, we introduce a spectral representation alignment approach to diffusion models, offer a geometric interpretation of why joint spectral learning benefits diffusion training, and establish its equivalence to diffusion score distillation in representation space. Integrating the resulting spectral regularizer into standard diffusion objectives yields consistent gains on image and 3D point cloud generation. These findings suggest a practical, principled path for further exploring the synergy between diffusion modeling and representation learning.

## THE USE OF LARGE LANGUAGE MODELS

Large language models were used solely for sentence-level proofreading and typographical correction. All research conception and manuscript writing were conducted by the authors.

## REPRODUCIBILITY STATEMENT

We include our experiment details in Section 5. Complete derivations and proofs are provided in Appendix B and Appendix C.

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

## A  PERTURBATION KERNELS OF CLASSIC DIFFUSION MODELS

**Rectified Flow.**  We show how to derive the forward SDE of rectified flow (Liu et al., 2022) from its perturbation kernel. Note that, the forward process in the original rectified flow is originally defined as $\boldsymbol{x}_t = (1-t)\boldsymbol{x}_0 + t\boldsymbol{\epsilon}$, $\boldsymbol{x}_0 \sim p_{\text{data}}$, $\boldsymbol{\epsilon} \sim \mathcal{N}(0, \boldsymbol{I})$. It appears this forward process is a linear interpolation between random noise and clean data samples, rather than in the form of SDE. In fact, it can be rewritten as an SDE using the perturbation kernel defined by the interpolation: $p_{0t}(\boldsymbol{x}_t|\boldsymbol{x}_0) = \mathcal{N}(\boldsymbol{x}_t; (1-t)\boldsymbol{x}_0, t\boldsymbol{I})$. Then, $s(t) = 1-t$, $\sigma(t) = \frac{t}{1-t}$. By Equation 2, $f(t) = -\frac{1}{1-t}$, $g(t) = \sqrt{\frac{2t}{1-t}}$. Then, we can write down the forward SDE as:

$$\mathrm{d}\boldsymbol{x} = -\frac{1}{1-t}\,\boldsymbol{x}\,\mathrm{d}t + \sqrt{\frac{2t}{1-t}}\mathrm{d}\boldsymbol{w}_t. \tag{17}$$

The corresponding reverse SDE is:

$$\mathrm{d}\boldsymbol{x} = \left[-\frac{1}{1-t}\,\boldsymbol{x} - \frac{2t}{1-t}\nabla_{\boldsymbol{x}}\log p_t(\boldsymbol{x})\right]\mathrm{d}t + \sqrt{\frac{2t}{1-t}}\mathrm{d}\boldsymbol{w}_t. \tag{18}$$

This SDE can be further converted into an ODE that preserves the marginal distribution $p_t(\boldsymbol{x})$:

$$\mathrm{d}\boldsymbol{x} = \underbrace{-\frac{1}{1-t}\left[\boldsymbol{x} + t\nabla_{\boldsymbol{x}}\log p_t(\boldsymbol{x})\right]\mathrm{d}t}_{\text{velocity field: } \boldsymbol{v}_t(\boldsymbol{x})}, \tag{19}$$

which yields the velocity field directly adopted in the original rectified flow approach. This relation between the score function and the velocity field in rectified flow is also shown in CFDM (Scarvelis et al., 2023).

## B  GEOMETRIC INTERPRETATION OF REPRESENTATIONS IN EIGENSPACE

In this section, we aim to give an interpretation of the representation learned from the diffusion process. Suppose all data points form a manifold $\mathcal{M}$. To find the similarity between data points on the manifold, diffusion distance is a metric measuring the probabilistic connectivity between

two data points via a random walk. Following the definition in Coifman & Lafon (2006), diffusion distance can be written as:

$$D_t^2(\boldsymbol{x}, \boldsymbol{x}') = \int_{\mathcal{M}} \left[ \frac{p_{0t}(\boldsymbol{x} \mid \boldsymbol{y})}{p_t(\boldsymbol{x})} - \frac{p_{0t}(\boldsymbol{x}' \mid \boldsymbol{y})}{p_t(\boldsymbol{x}')} \right]^2 p_0(\boldsymbol{y}) \, d\boldsymbol{y} \tag{20}$$

This diffusion distance is equivalent to the following one with respect to $\kappa_t(\boldsymbol{x}, \boldsymbol{x}')$:

$$D_{\kappa_t}^2(\boldsymbol{x}, \boldsymbol{x}') = \int_{\mathcal{M}} \left[ \kappa_t(\boldsymbol{x}, \boldsymbol{y}) - \kappa_t(\boldsymbol{x}', \boldsymbol{y}) \right]^2 p_t(\boldsymbol{y}) \, d\boldsymbol{y} \tag{21}$$

To see this, we can rewrite Equation 20 as:

$$D_t^2(\boldsymbol{x}, \boldsymbol{x}') = \int_{\mathcal{M}} \left[ \left( \frac{p_{0t}(\boldsymbol{x} \mid \boldsymbol{y})}{p_t(\boldsymbol{x})} \right)^2 + \left( \frac{p_{0t}(\boldsymbol{x}' \mid \boldsymbol{y})}{p_t(\boldsymbol{x}')} \right)^2 - 2 \frac{p_{0t}(\boldsymbol{x} \mid \boldsymbol{y}) p_{0t}(\boldsymbol{x}' \mid \boldsymbol{y})}{p_t(\boldsymbol{x}) p_t(\boldsymbol{x}')} \right] p_0(\boldsymbol{y}) d\boldsymbol{y} \tag{22}$$

$$= \int_{\mathcal{M}} \frac{p_{0t}^2(\boldsymbol{x} \mid \boldsymbol{y})}{p_t^2(\boldsymbol{x})} p_0(\boldsymbol{y}) d\boldsymbol{y} + \int_{\mathcal{M}} \frac{p_{0t}^2(\boldsymbol{x}' \mid \boldsymbol{y})}{p_t^2(\boldsymbol{x}')} p_0(\boldsymbol{y}) d\boldsymbol{y} - 2 \int_{\mathcal{M}} \frac{p_{0t}(\boldsymbol{x} \mid \boldsymbol{y}) p_{0t}(\boldsymbol{x}' \mid \boldsymbol{y})}{p_t(\boldsymbol{x}) p_t(\boldsymbol{x}')} p_0(\boldsymbol{y}) \, d\boldsymbol{y} \tag{23}$$

$$= \kappa_t(\boldsymbol{x}, \boldsymbol{x}) + \kappa_t(\boldsymbol{x}', \boldsymbol{x}') - 2\kappa_t(\boldsymbol{x}, \boldsymbol{x}') \tag{24}$$

Next, Equation 21 can be expanded into the following equation:

$$D_{\kappa_t}^2(\boldsymbol{x}, \boldsymbol{x}') = \int \kappa_t^2(\boldsymbol{x}, \boldsymbol{y}) p_t(\boldsymbol{y}) \, d\boldsymbol{y} + \int \kappa_t^2(\boldsymbol{x}', \boldsymbol{y}) p_t(\boldsymbol{y}) \, d\boldsymbol{y} - 2 \underbrace{\int \kappa_t(\boldsymbol{x}', \boldsymbol{y}) \kappa_t(\boldsymbol{x}, \boldsymbol{y}) p_t(\boldsymbol{y}) \, d\boldsymbol{y}}_{\mathcal{I}_1} \tag{25}$$

To show the equivalence, we begin with the simplification of the integral $\mathcal{I}_1$:

$$\mathcal{I}_1 := \int \kappa_t(\boldsymbol{x}', \boldsymbol{y}) \kappa_t(\boldsymbol{x}, \boldsymbol{y}) p_t(\boldsymbol{y}) \, d\boldsymbol{y} \tag{26}$$

$$= \int \frac{\int p_{0t}(\boldsymbol{x}'|\boldsymbol{w}) p_{0t}(\boldsymbol{y}|\boldsymbol{w}) p_0(\boldsymbol{w}) d\boldsymbol{w}}{p_t(\boldsymbol{x}') p_t(\boldsymbol{y})} \frac{\int p_{0t}(\boldsymbol{x}|\boldsymbol{u}) p_{0t}(\boldsymbol{y}|\boldsymbol{u}) p_0(\boldsymbol{u}) d\boldsymbol{u}}{p_t(\boldsymbol{x}) p_t(\boldsymbol{y})} p_t(\boldsymbol{y}) d\boldsymbol{y} \tag{27}$$

$$= \frac{1}{p_t(\boldsymbol{x}) p_t(\boldsymbol{x}')} \int \int p_{0t}(\boldsymbol{x}'|\boldsymbol{w}) p_{0t}(\boldsymbol{y}|\boldsymbol{w}) p_0(\boldsymbol{w}) d\boldsymbol{w} \int p_{0t}(\boldsymbol{x}|\boldsymbol{u}) p_{0t}(\boldsymbol{y}|\boldsymbol{u}) p_0(\boldsymbol{u}) d\boldsymbol{u} \frac{1}{p_t(\boldsymbol{y})} d\boldsymbol{y} \tag{28}$$

$$= \frac{1}{p_t(\boldsymbol{x}) p_t(\boldsymbol{x}')} \int \int \int p_{0t}(\boldsymbol{x}'|\boldsymbol{w}) p_{0t}(\boldsymbol{y}|\boldsymbol{w}) p_0(\boldsymbol{w}) p_{0t}(\boldsymbol{x}|\boldsymbol{u}) p_{0t}(\boldsymbol{y}|\boldsymbol{u}) p_0(\boldsymbol{u}) \frac{1}{p_t(\boldsymbol{y})} d\boldsymbol{w} d\boldsymbol{u} d\boldsymbol{y} \tag{29}$$

$$= \frac{1}{p_t(\boldsymbol{x}) p_t(\boldsymbol{x}')} \int \left[ p_{0t}(\boldsymbol{x}'|\boldsymbol{w}) p_0(\boldsymbol{w}) \right] \left[ p_{0t}(\boldsymbol{x}|\boldsymbol{u}) p_0(\boldsymbol{u}) \right] \underbrace{\left[ \int p_{0t}(\boldsymbol{y}|\boldsymbol{w}) p_{0t}(\boldsymbol{y}|\boldsymbol{u}) \frac{1}{p_t(\boldsymbol{y})} d\boldsymbol{y} \right]}_{\mathcal{I}_2} d\boldsymbol{w} d\boldsymbol{u} \tag{30}$$

In fact, the inner integral $\mathcal{I}_2$ is equal to $\frac{1}{p_0(\boldsymbol{w})} \delta(\boldsymbol{w} - \boldsymbol{u})$ by Bayes' rule:

$$\mathcal{I}_2 = \int \frac{p_{t0}(\boldsymbol{w}|\boldsymbol{y}) p_t(\boldsymbol{y}) p_{0t}(\boldsymbol{y}|\boldsymbol{u})}{p_0(\boldsymbol{w}) p_t(\boldsymbol{y})} d\boldsymbol{y} = \frac{1}{p_0(\boldsymbol{w})} \int p_{t0}(\boldsymbol{w}|\boldsymbol{y}) p_{0t}(\boldsymbol{y}|\boldsymbol{u}) d\boldsymbol{y} \tag{31}$$

By Chapman-Kolmogorov equation, we have:

$$\mathcal{I}_2 = \frac{1}{p_0(\boldsymbol{w})} \int p_{t0}(\boldsymbol{w}|\boldsymbol{y}) p_{0t}(\boldsymbol{y}|\boldsymbol{u}) d\boldsymbol{y} = \frac{1}{p_0(\boldsymbol{w})} p_{t \to t}(\boldsymbol{w}|\boldsymbol{u}) = \frac{1}{p_0(\boldsymbol{w})} \delta(\boldsymbol{w} - \boldsymbol{u}) \tag{32}$$

Then, we can substitute the simplified result of $\mathcal{I}_2$ to the Equation 30:

$$\mathcal{I}_1 = \frac{1}{p_t(\boldsymbol{x}) p_t(\boldsymbol{x}')} \int \left[ p_{0t}(\boldsymbol{x}'|\boldsymbol{w}) p_0(\boldsymbol{w}) \right] \left[ p_{0t}(\boldsymbol{x}|\boldsymbol{u}) p_0(\boldsymbol{u}) \right] \frac{1}{p_0(\boldsymbol{w})} \delta(\boldsymbol{w} - \boldsymbol{u}) d\boldsymbol{w} d\boldsymbol{u} \tag{33}$$

$$= \frac{1}{p_t(\boldsymbol{x}) p_t(\boldsymbol{x}')} \int \left[ p_{0t}(\boldsymbol{x}'|\boldsymbol{w}) p_0(\boldsymbol{w}) \right] \left[ p_{0t}(\boldsymbol{x}|\boldsymbol{w}) p_0(\boldsymbol{w}) \right] \frac{1}{p_0(\boldsymbol{w})} d\boldsymbol{w} \tag{34}$$

$$= \frac{1}{p_t(\boldsymbol{x}) p_t(\boldsymbol{x}')} \int p_{0t}(\boldsymbol{x}'|\boldsymbol{w}) p_{0t}(\boldsymbol{x}|\boldsymbol{w}) p_0(\boldsymbol{w}) d\boldsymbol{w} \tag{35}$$

$$= \frac{\mathbb{E}_{\boldsymbol{w}}[p_{0t}(\boldsymbol{x}'|\boldsymbol{w}) p_{0t}(\boldsymbol{x}|\boldsymbol{w})]}{p_t(\boldsymbol{x}) p_t(\boldsymbol{x}')} = \kappa_t(\boldsymbol{x}, \boldsymbol{x}') \tag{36}$$

The other two integrals in Equation 25 can be treated as special cases of $\mathcal{I}_1$. Thus, we can finally approach the desired equivalence of Equation 20 and Equation 21:

$$D_{\kappa_t}^2(\boldsymbol{x}, \boldsymbol{x}') = \int_{\mathcal{M}} \left[\kappa_t(\boldsymbol{x}, \boldsymbol{y}) - \kappa_t(\boldsymbol{x}', \boldsymbol{y})\right]^2 p_t(\boldsymbol{y}) \, d\boldsymbol{y} \tag{37}$$

$$= \kappa_t(\boldsymbol{x}, \boldsymbol{x}) + \kappa_t(\boldsymbol{x}', \boldsymbol{x}') - 2\kappa_t(\boldsymbol{x}, \boldsymbol{x}') = D_t^2(\boldsymbol{x}, \boldsymbol{x}') \tag{38}$$

By Mercer's theorem, since $\kappa_t(\boldsymbol{x}, \boldsymbol{x}')$ is symmetric and positive definite, we have the following expansion of $\kappa_t(\boldsymbol{x}, \boldsymbol{x}')$:

$$\kappa_t(\boldsymbol{x}, \boldsymbol{x}') = \sum_{l=0}^{\infty} \lambda_{t,l} \psi_{t,l}(\boldsymbol{x}) \psi_{t,l}(\boldsymbol{x}'), \tag{39}$$

where $\psi_{t,l}(\boldsymbol{x})$ is the $l$-th eigenfunction of the integral operator $\mathcal{K}_t$. Note that $\{\psi_{t,l}(\boldsymbol{x})\}_l$ is a set of orthonormal functions, where $\psi_{t,l}(\boldsymbol{x})$ is corresponding to the $l$-th largest eigenvalue $\lambda_{t,l}(\boldsymbol{x})$:

$$\delta_{lm} = \int \psi_{t,l}(\boldsymbol{w}) \psi_{t,m}(\boldsymbol{w}) p_t(\boldsymbol{w}) d\boldsymbol{w} = \begin{cases} 1, & l = m, \\ 0, & l \neq m \end{cases}. \tag{40}$$

We can further use this set of orthonormal eigenfunctions to represent the diffusion distance:

$$D_t^2(\boldsymbol{x}, \boldsymbol{x}') = \int \left[\kappa_t(\boldsymbol{x}, \boldsymbol{w}) - \kappa_t(\boldsymbol{x}', \boldsymbol{w})\right]^2 p_t(\boldsymbol{w}) \, d\boldsymbol{w} \tag{41}$$

$$= \int \left[\sum_{l=0}^{\infty} \lambda_{t,l} \psi_{t,l}(\boldsymbol{x}) \psi_{t,l}(\boldsymbol{w}) - \sum_{m=0}^{\infty} \lambda_{t,m} \psi_{t,m}(\boldsymbol{x}') \psi_{t,m}(\boldsymbol{w})\right]^2 p_t(\boldsymbol{w}) \, d\boldsymbol{w} \tag{42}$$

$$= \int \left[\sum_{l=0}^{\infty} \lambda_{t,l} \left(\psi_{t,l}(\boldsymbol{x}) - \psi_{t,l}(\boldsymbol{x}')\right) \psi_{t,l}(\boldsymbol{w})\right]^2 p_t(\boldsymbol{w}) \, d\boldsymbol{w} \tag{43}$$

$$= \int \left[\sum_{l,m=0}^{\infty} \lambda_{t,l} \lambda_{t,m} \left[\psi_{t,l}(\boldsymbol{x}) - \psi_{t,l}(\boldsymbol{x}')\right] \left[\psi_{t,m}(\boldsymbol{x}) - \psi_{t,m}(\boldsymbol{x}')\right] \psi_{t,l}(\boldsymbol{w}) \psi_{t,m}(\boldsymbol{w}) p_t(\boldsymbol{w})\right] d\boldsymbol{w} \tag{44}$$

$$= \sum_{l,m=0}^{\infty} \lambda_{t,l} \lambda_{t,m} \left[\psi_{t,l}(\boldsymbol{x}) - \psi_{t,l}(\boldsymbol{x}')\right] \left[\psi_{t,m}(\boldsymbol{x}) - \psi_{t,m}(\boldsymbol{x}')\right] \int \psi_{t,l}(\boldsymbol{w}) \psi_{t,m}(\boldsymbol{w}) p_t(\boldsymbol{w}) d\boldsymbol{w} \tag{45}$$

$$= \sum_{l,m=0}^{\infty} \lambda_{t,l} \lambda_{t,m} \left[\psi_{t,l}(\boldsymbol{x}) - \psi_{t,l}(\boldsymbol{x}')\right] \left[\psi_{t,m}(\boldsymbol{x}) - \psi_{t,m}(\boldsymbol{x}')\right] \delta_{lm} \tag{46}$$

$$= \sum_{l=0}^{\infty} \lambda_{t,l}^2 \left[\psi_{t,l}(\boldsymbol{x}) - \psi_{t,l}(\boldsymbol{x}')\right]^2 \tag{47}$$

By constructing the first $K$ eigenfunctions as an embedding: $\boldsymbol{\xi}_t(\boldsymbol{x}) = [\lambda_{t,0} \psi_{t,0}(\boldsymbol{x}), ..., \lambda K \psi_{t,K}(\boldsymbol{x})]$, the L2 distance between $\boldsymbol{\xi}_t(\boldsymbol{x})$ and $\boldsymbol{\xi}_t(\boldsymbol{x}')$ approximates the diffusion distance between $\boldsymbol{x}$ and $\boldsymbol{x}'$ on the manifold evolved at $t$. Therefore, applying Neural Eigenmap objectives to regularize diffusion model training can be interpreted as

enforcing time-evolving geometric structure on the intermediate hidden states of networks. This geometric regularization guides the model to denoise data with varying perturbations in a consistent manner, which is expected to alleviate the training challenges in diffusion models.

## C   DUALITY OF SPECTRAL REPRESENTATION LEARNING AND CLOSED-FORM DIFFUSION SCORE DISTILLATION

We adopt the result of Garrido et al. (2022) that dimension-contrastive and sample-contrastive self-supervised objectives are equivalent when representation embeddings are normalized across chan-

nels and mini-batches. The spectral regularization can finally have this equivalent form:

$$\min_\theta - \sum_{i=1}^{B} \psi_\theta(\boldsymbol{x}_i, t)^\top \psi_\theta(\boldsymbol{x}_i', t) + \sum_{i=1}^{B} \sum_{j \neq i} \psi_\theta(\boldsymbol{x}_i, t)^\top \psi_\theta(\boldsymbol{x}_j, t) \tag{48}$$

$$\Leftrightarrow \min_\theta - \sum_{i=1}^{B} \left( \frac{\psi_\theta(\boldsymbol{x}_i, t)^\top \psi_\theta(\boldsymbol{x}_i', t)}{\tau} \right) + \sum_{i=1}^{B} \log \left[ \sum_{j \neq i} \exp \left( \frac{\psi_\theta(\boldsymbol{x}_i, t)^\top \psi_\theta(\boldsymbol{x}_j, t)}{\tau} \right) \right], \tag{49}$$

where $\tau$ denotes a temperature hyperparameter. As the spectral embedding $\psi(\boldsymbol{x}_i, t)$ is normalized, the above optimization problem can be further re-written as the following one:

$$\min_\theta \underbrace{- \sum_{i=1}^{B} \log \left[ \exp \left( \frac{-\|\psi_\theta(\boldsymbol{x}_i, t) - \psi_\theta(\boldsymbol{x}_i', t)\|_2^2}{\tau} \right) \right]}_{:=\mathcal{L}_s^+} \tag{50}$$

$$+ \underbrace{\sum_{i=1}^{B} \log \left[ \sum_{j \neq i} \exp \left( \frac{-\|\psi_\theta(\boldsymbol{x}_i, t) - \psi_\theta(\boldsymbol{x}_j, t)\|_2^2}{\tau} \right) \right]}_{:=\mathcal{L}_s^-}, \tag{51}$$

where we transform the dot product operations to L2 distance. Interestingly, when $\psi_\theta(\boldsymbol{x}_j, t)$ in $\mathcal{L}_s^-$ and $\psi_\theta(\boldsymbol{x}_i', t)$ in $\mathcal{L}_s^+$ are detached from gradient propagation (which is true in our adopt NeuralEF (Deng et al., 2022b) approach), their derivatives regarding $\psi_\theta(\boldsymbol{x}_i, t)$ are in the similar form of batch-wise closed-form score of diffusion models in the representation embedding space:

$$\nabla_{\psi_\theta(\boldsymbol{x}_i, t)} \mathcal{L}_s^+ = \frac{2}{\tau} \left( \psi_\theta(\boldsymbol{x}_i, t) - \psi_\theta(\boldsymbol{x}_i', t) \right) \tag{52}$$

$$\nabla_{\psi_\theta(\boldsymbol{x}_i, t)} \mathcal{L}_s^- = \frac{2}{\tau} \sum_{k \neq i} \frac{\exp \left( -\|\psi_\theta(\boldsymbol{x}_i, t) - \psi_\theta(\boldsymbol{x}_k, t)\|_2^2 / \tau \right)}{\sum_{j \neq i} \exp \left( -\|\psi_\theta(\boldsymbol{x}_i, t) - \psi_\theta(\boldsymbol{x}_j, t)\|_2^2 / \tau \right)} \left( \psi_\theta(\boldsymbol{x}_k, t) - \psi_\theta(\boldsymbol{x}_i, t) \right) \tag{53}$$

The gradient expressions in Equation 53 and 52 resemble the closed-form score of diffusion models (Scarvelis et al., 2023). Given a training set $\mathcal{D} = \{\boldsymbol{x}_i\}_{i=0}^{D}$ with $D$ samples, the closed-form expression of the score function under the rectified flow formulation can be written as:

$$\nabla_{\boldsymbol{z}} \log p_t(\boldsymbol{z}) = \frac{1}{t^2} \sum_{k=1}^{D} \frac{\exp \left( -\|\boldsymbol{z} - (1 - t)\boldsymbol{x}_k\|_2^2 / 2t^2 \right)}{\sum_{j=1}^{D} \exp \left( -\|\boldsymbol{z} - (1 - t)\boldsymbol{x}_j\|_2^2 / 2t^2 \right)} \left( (1 - t)\boldsymbol{x}_k - \boldsymbol{z} \right), \tag{54}$$

where $\boldsymbol{z} = (1 - t)\boldsymbol{x} + t\boldsymbol{\epsilon}$, $\boldsymbol{x} \sim \mathcal{D}$, $\boldsymbol{\epsilon} \sim \mathcal{N}(0, I)$, $\forall t \in (0, 1]$. By comparing equations 54 and 53: the temperature $\tau$ can be seen as $2t^2$, the counterparts of $\psi_\theta(\boldsymbol{x}_k, t)$ in the numerator and $\psi_\theta(\boldsymbol{x}_j, t)$ in the denominator are $(1 - t)\boldsymbol{x}_k$ and $(1 - t)\boldsymbol{x}_j$, and data samples for evaluating the gradient in Equation 53 are those negative samples. The notation in Equation 52 is defined analogously; the difference is that the score is evaluated at a single positive sample.

In this sense, the total derivative $\partial \mathcal{L}_s / \partial \psi_\theta(\boldsymbol{x}_i, t) = \nabla_{\psi_\theta(\boldsymbol{x}_i, t)} \mathcal{L}_s^+ + \nabla_{\psi_\theta(\boldsymbol{x}_i, t)} \mathcal{L}_s^-$ is a score function evaluated on a sampled data batch. Intuitively, $\nabla_{\psi_\theta(\boldsymbol{x}_i, t)} \mathcal{L}_s^-$ points at the direction which is a weighted sum of displacement vectors from $\psi_\theta(\boldsymbol{x}_i, t)$ to $\psi_\theta(\boldsymbol{x}_k, t)$ for all $k \neq i, k \in [B]$. The pairwise weights decrease with the squared L2 distances and are normalized by the softmax function. Once $\psi_\theta$ is learned to represent eigenfunctions, the displacement vectors are weighted by the diffusion distance (without eigenvalue weighting) of data samples (see Appendix **??**). Conversely, $\nabla_{\psi_\theta(\boldsymbol{x}_i, t)} \mathcal{L}_s^+$ points away from the positive sample's representation $\psi_\theta(\boldsymbol{x}_i', t)$, akin to the negative-prompting in diffusion models.

Next, we can show that optimizing our spectral regularization term is actually conducting a score distillation. For $\boldsymbol{x} \sim p_t(\boldsymbol{x})$, $\psi_\theta(\cdot, t)$ can be seen as a generator: $\psi_\theta(\boldsymbol{x}, t) \sim p_t^{\psi_\theta}$, where $p_t^{\psi_\theta}$ is a latent distribution of spectral embeddings. A score distillation step from $p_t^{\psi_\theta}$ to a target distribution $p_{\text{target}}$ can be achieved by minimizing their KL divergence through a gradient-based optimizer. Specifically, the gradient of KL divergence w.r.t $\theta$ is:

$$\nabla_\theta D_{\text{KL}}(p_t^{\psi_\theta} \| p_{\text{target}}) = \mathbb{E}_{\boldsymbol{x} \sim p_t} \left[ (\nabla_\theta \psi_\theta(\boldsymbol{x}, t))^\top \left( \nabla_{\psi_\theta(\boldsymbol{x}, t)} \log p_t^{\psi_\theta} - \nabla_{\psi_\theta(\boldsymbol{x}, t)} \log p_{\text{target}} \right) \right] \tag{55}$$

Let $p_{\text{target}}$ be a Gaussian mixture centered at positive samples with bandwidth $\tau$ (in our case, there is only one positive sample), and model the latent distribution $p_t^{\psi_\theta}$ as a Gaussian mixture over negative samples with the same bandwidth $\tau$, we have $\nabla_{\psi_\theta(\boldsymbol{x},t)} \log p_t^{\psi_\theta} = \nabla_{\psi_\theta(\boldsymbol{x}_i,t)} \mathcal{L}_s^-$ and $\nabla_{\psi_\theta(\boldsymbol{x},t)} \log p_{\text{target}} = -\nabla_{\psi_\theta(\boldsymbol{x}_i,t)} \mathcal{L}_s^+$.

Therefore, the gradient of the score distillation step turns out to be:

$$\nabla_\theta D_{\text{KL}}(p_t^{\psi_\theta} \parallel p_{\text{target}}) = \mathbb{E}_{\boldsymbol{x} \sim p_t} \left[ (\nabla_\theta \psi_\theta(\boldsymbol{x}, t))^\top \left( \nabla_{\psi_\theta(\boldsymbol{x},t)} \mathcal{L}_s^- + \nabla_{\psi_\theta(\boldsymbol{x},t)} \mathcal{L}_s^+ \right) \right] \qquad (56)$$

$$= \mathbb{E}_{\boldsymbol{x} \sim p_t} \left[ (\nabla_\theta \psi_\theta(\boldsymbol{x}, t))^\top \nabla_{\psi_\theta(\boldsymbol{x},t)} \mathcal{L}_s \right] \qquad (57)$$

By the chain rule, the gradient of the original spectral representation objective w.r.t $\theta$ is:

$$\frac{\partial \mathcal{L}_s}{\partial \theta} = \mathbb{E}_{\boldsymbol{x} \sim p_t} \left[ (\nabla_\theta \psi_\theta(\boldsymbol{x}, t))^\top \nabla_{\psi_\theta(\boldsymbol{x},t)} \mathcal{L}_s \right] \equiv \nabla_\theta D_{\text{KL}}(p_t^{\psi_\theta} \parallel p_{\text{target}}) \qquad (58)$$

This concludes the proof that shows optimizing the spectral representation regularizer is performing diffusion score distillation.

