# OpenReview forum: "Revisiting Spectral Representations in Generative Diffusion Models"
_ICLR.cc/2026/Conference — Submitted to ICLR 2026_

### Official Review · Reviewer_A9rz · 2025-10-29

**Soundness:** 3
**Presentation:** 1
**Contribution:** 2
**Rating:** 4
**Confidence:** 3

**Summary:**

The authors propose a spectral regularizer for diffusion models with a projection head trained to approximate the top-K eigenfunctions of the forward-diffusion kernel. The authors link the resulting objective to score-distillation in representation space and report gains on several visual and point cloud datasets.

**Strengths:**

The method proposed is teacher-free, and the authors connect it to the forward-diffusion kernel, which is good.
The implementation should be fairly efficient, with a time-conditioned head.
The data ablations considered are reasonable for synthetic data, images, and point clouds.
The results seem to yield mild but consistent gains, especially on point clouds or low res images.

**Weaknesses:**

The reported results are severely lacking in statistical significance. For all trend figures and tables the authors should (at the very least) report average and std/err over three or more seeds. Because of the mild improvements in some cases, and the lack of multi-run statistics, it is hard to judge significance.

The results are limited to mostly low-resolution experiments, and there seems to be a negative correlation in Table 1 between FID improvements and scale.

**Questions:**

1. It is hard to judge the significance of the reported results due to the lack of reported statistics over multiple runs. Can you report mean+std error for each of the metric figure/tables in the paper?

2. It would be worthwhile to understand the reason behind the diminishing FID improvements in table 1. Are these due to scale, or perhaps working with latents on ImageNet? Statistics over multiple runs could also help here, as well as additional latent diffusion experiments time allowing.

---

> ### Author Response · Authors · 2025-12-03
>
> * **Reporting mean $\pm$ std.**
> It is standard practice in diffusion-model evaluation *not* to report variance for FID or related metrics. Prior works such as REPA, DiT, and SiT also omit mean $\pm$ std, because these metrics are computed over 50k generated samples, in which statistical tests may provide limited additional insight. More importantly, generating 50k images for each evaluation run requires hours of computation, making multiple independent runs infeasible in practice. Thus, we just follow established convention in diffusion evaluation.
>
> * **Limited FID improvements on large datasets.**
> Our method shows stronger gains on smaller datasets, which is consistent with our setting: SRL is designed for scenarios where data is limited and pretrained encoders are *not* available. On large-scale datasets, where foundation encoders can be trained and used for representation alignment (as in REPA), their advantage naturally becomes more pronounced. Our method remains useful in domains where such encoders cannot be obtained on large-scale datasets.

---

### Official Review · Reviewer_CYG5 · 2025-10-31

**Soundness:** 3
**Presentation:** 2
**Contribution:** 2
**Rating:** 2
**Confidence:** 4

**Summary:**

This paper explores the intrinsic link between self-supervised Spectral Representation Learning (SRL) and Diffusion Models through the shared lens of time-dependent perturbation kernels. The authors propose a novel self-supervised spectral regularization loss to align the intermediate representations of the diffusion network, successfully avoiding the need for external pre-trained encoders. Empirical results consistently demonstrate that the proposed method significantly enhances the performance across both image and 3D point cloud synthesis tasks.

**Strengths:**

- The method offers a novel and versatile self-supervised regularization, requiring no external pre-trained encoder.
- The proposed method is applicable across multiple modalities and resolutions.
- Solid theoretical analysis connects the spectral loss to the mode-seeking dynamics of score distillation.

**Weaknesses:**

- Sensitivity analysis for the critical regularization hyperparameter $\lambda$ is insufficient.
- There lacks a discussion of the specific intermediate hidden layer for alignment.
- The improvements in FID scores, while consistent, are relatively modest.

**Questions:**

1. **Ablation on simplest contrastive loss.** Despite sufficient motivation explanations and theoretical analysis, practically, I believe the proposed SRL factually acts like a consistent loss across different perturbation levels. A similar line of this work is to use the simplest contrastive loss (w/ or w/o negative samples) to align the intermediate features across different noises. Could the authors provide an ablation study on this simplest contrastive loss to compare with the proposed SRL, to better justify the effectiveness of SRL?

2. **The improvement is limited.** Despite without using any external pre-trained encoder, the improvement on FID seems to be insufficient to demonstrate the effectiveness of the proposed method, especially that on CelebA. Could the authors provide more comprehensive experiments including:
   - combining SRL with REPA and reporting the comparision
   - similar to REPA, showing how many times can SRL boost training

3. **Training curve.** Could the authors explain why the performance of SRL is worse than the baseline at the beginning of training on ImageNet-64 in Figure 2?

4. **Sensitivity of $\lambda$.** The regularization weight $\lambda$ is critical to the final performance. Could the authors provide a sensitivity analysis on $\lambda$ to show how it affects the final performance?

5. **Ablation on the hidden layer.** Could the authors provide an ablation study on which intermediate hidden layer to apply the proposed SRL? How does it affect the final performance?

---

> ### Author Response · Authors · 2025-12-03
>
> * **Ablation on hidden layer.**
>   We performed a brief ablation on the choice of hidden layer where the regularizer is applied. The results on CIFAR-10 are shown below. The variations are small, indicating that our method is not highly sensitive to the specific layer selected.
>
> | Layer |  L=2  |  L=3  |   L=5 |
> | :---- | :---: | :---: | ----: |
> | FID   | 8.613 | 9.073 | 8.742 |
>
> * **Ablation on $\lambda$.**
>   We also conducted an ablation on the regularization weight $\lambda$, using CIFAR10 dataset. As shown below, the method remains stable across a range of values. Importantly, our reported experiments do not tune $\lambda$ for optimal results; they instead demonstrate the general effectiveness and robustness of our approach.
>
> | $\lambda$ |  0.05 |  0.5  |  0.1  |
> | :-------: | :---: | :---: | :---: |
> |    FID    | 8.742 | 8.578 | 8.628 |
>
> * **Training curve explanation.**
>   Regarding the observation that SRL underperforms the baseline at early training stages: this occurs because spectral representations are initially unrefined and thus provide limited guidance. After several thousand iterations, the spectral learner becomes meaningful, and the regularizer begins contributing positively, leading to the consistent improvements observed later in training.
>
> * **Combining SRL and REPA losses.**
>   We appreciate the reviewer's suggestion and will explore this combination. Nonetheless, the primary focus of the paper is on self-supervised representation alignment for data modalities where pretrained encoders are unavailable.

---

### Official Review · Reviewer_R6Ra · 2025-10-31

**Soundness:** 2
**Presentation:** 2
**Contribution:** 2
**Rating:** 2
**Confidence:** 4

**Summary:**

This paper adds an additional self supervised alignment loss to the standard denoising score matching objective, so that similar samples in the representation space are expected to be clustered while different samples are pushed apart.

**Strengths:**

Adding extra self supervised regularization to the representation space for training diffusion models is interesting.

**Weaknesses:**

1. The motivation of the proposed alignment method is unclear. What is rationale behind minimizing the KL divergence term in (16)? Why do we want the score of negative samples to match the score of positive samples? Why is this beneficial?

2. The performance gain from the proposed algorithm is marginal. In particular, in table 1, very often, the metrics only marginally improved. From my experience, such marginal improvement won't affect the perceptual quality of the generated samples, and I cannot find any qualitative results for image generation in the paper.

**Questions:**

1. I don't understand the reasoning in line 305-310. Why minimizing the KL divergence leads to mode seeking behavior, clustering similar data while pushing apart dissimilar ones? Please explain.

2. The performance gain is marginal, does your method improve image quality at all?

3. Have you tried CFG? How does the model performance change when you apply CFG? Please plot the FID v.s. Inception curve for varying guidance strength for both the base model and your model.

---

> ### Author Response · Authors · 2025-12-03
>
> * **Motivation of the proposed alignment.**
>   Our alignment mechanism is motivated by two complementary insights.
>   1) Spectral alignment encourages representations to respect diffusion distance, thereby encoding meaningful relationships between images. When two images are close under diffusion distance, their hidden states are encouraged to be similar. This provides geometric guidance during denoiser training.
>   2) The spectral regularizer behaves as an auxiliary diffusion distillatiom loss that pushes representations toward in-cluster distributions. This encourages similar images to share similar hidden states, improving consistency of the denoiser's predictions for similar images.
>
> * **Rationale for mode-seeking behavior of the KL divergence.**
>   When minimizing the KL divergence, only the positive samples' representations ($\psi_\theta(\mathbf{x}_t^+)$) are updated. The gradient points toward a weighted mean of nearby negative representations, effectively pulling each positive embedding toward clusters of negatives that share similar features. Because positive samples correspond to noisy versions of the same image, while negatives may include images with partial similarity, the regularizer encourages clustering of similar representations. This mode-seeking behavior facilitates more consistent velocity (or clean-image) predictions for similar images.
>
> * **Performance gains.**
>   While the improvements on image datasets vary across settings, they are consistent and non-trivial. Notably, on CIFAR-10 and CelebA, FID improves by roughly 3 points, which are significant by diffusion model standards. Moreover, the gains on point-cloud generation are even larger, showing the effectiveness of our approach in domains without pretrained encoders.
>
> * **Use of CFG.**
>   We use CFG during inference and follow the default coefficients from the public codebase to ensure a fair comparison; we do not tune CFG specifically for our method. We will consider adding a plot of FID versus guidance strength to illustrate CFG's effect, although we view this as supplemental, since image generation is not the sole focus of our experiments.

---

### Official Review · Reviewer_xxCz · 2025-11-01

**Soundness:** 3
**Presentation:** 4
**Contribution:** 3
**Rating:** 6
**Confidence:** 3

**Summary:**

The paper proposes a method that applies spectral representation learning to diffusion models. To do this, additional MLP projection head is attached for learning spectral embedding and a spectral loss is added to train the embedding and this works as a regularizer to encourage representation alignment between perturbed samples derived from the same clean data. This can be seen as replacing the perturbation kernel used in traditional spectral representation learning with the diffusion noise kernel. Without any external model like REPA, the model could achieve better performance than baseline models on both ImageNet image generation and 3D point-cloud generation tasks.

**Strengths:**

- The paper proposes a novel connection between spectral representation learning and diffusion models, providing new insight by viewing them under a unified framework.
- The paper is well-written, with clear explanations of the preliminary background and methodology, which enhances readability.
- The method appears relatively easy to implement and improves internal representations without relying on any external models.
- The approach demonstrates noticeable performance improvements across very different domains, including ImageNet and 3D point cloud generation.

**Weaknesses:**

- The comparison experiments are somewhat limited. Although REPA is included, comparisons with other feature regularization methods such as Dispersive Loss are missing.
- While the method does not require external models, its generation quality appears lower than REPA, raising questions about its practical usefulness in real applications.
- In Figure 2, unlike REPA, the method does not seem to converge faster or show any clear trend in training behavior. To confirm that the improvement is not due to random fluctuations, multiple runs with statistical analysis would be necessary.

**Questions:**

- How much additional computational cost is introduced by adding the spectral branch?
- Are there any visualization comparisons for the image generation results?
- Is f in L133 a typo of h?

---

> ### Author Response · Authors · 2025-12-03
>
> * **Performance lower than REPA.**
>   Our method is designed for settings *without* access to pretrained encoders, such as CelebA or point-cloud datasets. REPA leverages large-scale pretrained encoders (e.g., DINO/CLIP), which provide substantial external supervision. For ImageNet-256, we therefore view REPA's performance as an approximate upper bound, whereas our approach operates under a strictly weaker assumption.
>
> * **On visual comparisons.**
>   Uncurated image comparisons often exhibit subtle differences that are difficult to interpret reliably. For this reason, we primarily rely on quantitative metrics to assess generative quality, which is consistent with common practice in diffusion-model evaluation.
>
> * **On statistical analysis.**
>   Prior diffusion papers, including REPA, DiT, and SiT, typically do not report variances for FID or related metrics. This is because these metrics are computed using 50k generated samples, in which the generation quality evaluation could be considered stable. In addition, generating the 50k samples needed per evaluation run takes hours, so performing multiple runs is not practically feasible. Thus, we follow this established convention in diffusion model evaluation.
>
> * **Additional computational cost.**
>   Incorporating the spectral branch increases the training cost to approximately 1.7× that of the baseline.
>
> * We thank the reviewer for catching this typo; we will correct it in the revised manuscript.

---

### Meta-Review · Area_Chair_k2gT · 2026-01-07

**Summary:**

The initial review scores for this paper is quite negative: 2,2,4,6.

Major concerns raised by reviewers include: comparisons with other feature regularization methods such as Dispersive Loss are missing; generation quality appears lower than REPA; marginal performance gains; motivation like the proposed alignment method; clarity of some technical parts; lack of ablation studies; the results are limited to mostly low-resolution experiments;

**Reviewer Concerns:**

Concerns like ablation studies, technical clarity and motivation seems to be addressed to some extent.

But several major concerns still remains:
--comparisons with other feature regularization methods such as Dispersive Loss are missing;
--marginal performance gains;
--the results are limited to mostly low-resolution experiments;

**Reviewer Scores:**

Some of the reviewer scores 2,2,4 might be increased, but I don't think the average scores will be raised to "accept".

---

### Decision · Program_Chairs · 2026-01-26

Reject